# Specialized 16SrX phytoplasmas induce diverse morphological and physiological changes in their respective fruit crops

**Jannicke Gallinger**[1], **Kerstin Zikeli**[1], **Matthias R. Zimmermann**[2], **Louisa M. Görg**[1], **Axel Mithöfer**[3], **Michael Reichelt**[4], **Erich Seemüller**[1], **Jürgen Gross**[1], **Alexandra C. U. Furch**[2]*

**1** Institute for Plant Protection in Fruit Crops and Viticulture, Julius Kühn-Institut, Federal Research Institute for Cultivated Plants, Dossenheim, Germany, **2** Plant Physiology, Matthias-Schleiden-Institute for Genetics, Bioinformatics and Molecular Botany, Faculty of Biological Science, Friedrich-Schiller-University Jena, Jena, Germany, **3** Research Group Plant Defense Physiology, Max-Planck Institute for Chemical Ecology, Jena, Germany, **4** Department of Biochemistry, Max-Planck Institute for Chemical Ecology, Jena, Germany

* Alexandra.furch@uni-jena.de

**Data Availability Statement:** All relevant data are within the manuscript and its Supporting Information files.

## Abstract

The host-pathogen combinations—*Malus domestica* (apple)/'*Candidatus* Phytoplasma mali´, *Prunus persica* (peach)/'*Ca*. P. prunorum´ and *Pyrus communis* (pear)/'*Ca*. P. pyri´ show different courses of diseases although the phytoplasma strains belong to the same 16SrX group. While infected apple trees can survive for decades, peach and pear trees die within weeks to few years. To this date, neither morphological nor physiological differences caused by phytoplasmas have been studied in these host plants. In this study, phytoplasma-induced morphological changes of the vascular system as well as physiological changes of the phloem sap and leaf phytohormones were analysed and compared with non-infected plants. Unlike peach and pear, infected apple trees showed substantial reductions in leaf and vascular area, affecting phloem mass flow. In contrast, in infected pear mass flow and physico-chemical characteristics of phloem sap increased. Additionally, an increased callose deposition was detected in pear and peach leaves but not in apple trees in response to phytoplasma infection. The phytohormone levels in pear were not affected by an infection, while in apple and peach trees concentrations of defence- and stress-related phytohormones were increased. Compared with peach and pear trees, data from apple suggest that the long-lasting morphological adaptations in the vascular system, which likely cause reduced sap flow, triggers the ability of apple trees to survive phytoplasma infection. Some phytohormone-mediated defences might support the tolerance.

## Author summary

Tons of fruits get lost each year by pathogenic phytoplasma infections of stone and pome fruit trees worldwide. Besides clearing infected trees, no effective control strategies are available and more specific studies are mandatory to enhance pest managements. Whereas phytoplasma genome sequencing has stimulated the deciphering of molecular

**Funding:** ACUF and MRZ was supported by the Deutsche Forschungsgemeinschaft (www.dfg.de) (Grant FU 969/2-1). JaG was supported by a fund of the "Landwirtschaftliche Rentenbank" (www. rentenbank.de) number 28RF4IP008. The funders had no role in study design, data collection and analysis, decision to publish, or preparation of the manuscript.

**Competing interests:** The authors have declared that no competing interests exist.

mechanisms, the process of physiological and morphological responses of host plants is poorly understood. This is the first comprehensive and precise study on the influence of specific phytoplasma species on the vascular system and leaf phytohormone levels of their Rosaceae host plants (apple, peach and pear) at the same time. This study reveals anatomical and physiological changes in apple trees, which may result from an enhanced ability to sense phytoplasma infection and provoke adequate and successful cellular and physiological defence responses. In contrast, peach and pear trees seem to miss this ability, causing the often-observed fast death of infected trees. The study provides new and important insights into the phytoplasma-induced diseases and symptom development. Improved understanding of the diseases is vital for the development of sustainable pest management strategies and breeding resistant cultivars.

## Introduction

The fruit tree diseases apple proliferation (AP), pear decline (PD) and European stone fruit yellows (ESFY), are of high economic significance, causing annual crop losses of around half a billion Euro in Europe, alone [1,2]. Intra- and interspecific differences in the response of fruit trees to these phytoplasma diseases have been observed over the last decades under both experimental and natural infection conditions [3,4]. Intraspecific differences have been explained by varying susceptibility of tree species and genotypes (rootstocks and cultivars) to phytoplasmas as well as virulence of phytoplasma strains [5–9]. However, only few studies provide firm data on host response, host–pathogen interaction and on anatomical, physiological and molecular basis of plant resistance [10], which is still poorly understood [4]. While the host plants of the 16SrX phytoplasmas belong to the Rosaceae, the causing agents of the diseases 'Candidatus Phytoplasma mali', 'Candidatus Phytoplasma pyri' and 'Candidatus Phytoplasma prunorum' are also phylogenetically closely related [11]. Phytoplasmas are very small bacteria without a cell wall. As all bacteria including phytoplasmas have generally circular chromosomes, the 16SrX phytoplasmas used in this study, *Ca.* P. mali, *Ca.* P. pyri and *Ca.* P. prunorum, have small linear chromosomes. Phytoplasmas lack many genes that encode important metabolic functions such as amino and fatty acid synthesis [12,13] and therefore, they need to consume essential metabolites from their plant hosts.

Phytoplasmas are restricted to the phloem sieve elements in their host plants [14,15]. The phloem serves as main route for the long and short-distance transport of mainly organic compounds [16,17]. Sieve elements (SEs), companion cells and phloem parenchyma cells are the three phloem cell types involved also in transport of defence- and stress related signalling molecules, such as RNA, proteins, and phytohormones e.g. [18–21]. The sieve element sap is an energy-rich environment, sustaining phytoplasmas with nutrients and enabling them to distribute all over the plant. Therefore, upon phytoplasma infection the impairment of the phloem cells and the change in the phloem sap composition are expectable.

The distribution of secondary compounds plays a crucial role in plant communication and the induction of defence mechanisms against invading pathogens and attacking herbivores. It was previously shown that phytoplasmas produce and secrete effector proteins into phloem sap that circulate to distal tissues and induce physiological changes in infected host plants [22]. A number of non-specific symptoms, such as chlorosis, leaf yellowing, premature reddening, swollen leaf-veins, leaf curl and reduced vigour might be attributed to the impairment of the vascular system and the photosynthesis apparatus [23,24]. Additionally, abnormal growth, stunting, growth of witches' brooms, reduced root size and dwarf fruits occur in phytoplasma

infected plants indicating a disturbed hormone balance [25,26]. Phytohormones are induced in reaction to abiotic and biotic stresses and lead to the induction of defence responses [27]. The influence of phytoplasma infections on salicylic acid, jasmonates, auxins, abscisic acid, ethylene and cytokinine biosynthesis and pathways was recently reviewed by Dermastia [26], illustrating the diverse and complex interactions between the specialized pathogens and their host plants.

In the case of phytoplasmas, the impact on vector insects that are crucial for the distribution of phytoplasmas, has to be taken into consideration. So far, all phytoplasmas of the group 16SrX causing important fruit crop diseases are vectored by jumping plant lice (Hemiptera: Psylloidea) or succinctly psyllids [28–30]. Psyllid nymphs and adults feed on plant phloem and occasionally on xylem sap [31–33]. Therefore, morphological changes of the plant vascular system may affect psyllid feeding behaviour and suitability of plants as hosts of vector insects. Additionally, phloem/xylem components may influence host choice and oviposition behaviour of psyllids [31,32,34,35]. To detect appropriate host plants for feeding and reproduction, volatile signals are used by many vectoring psyllid species during migration [36–43]. As plant volatile emission is frequently regulated by phytohormones, their changes in concentrations play an important role on the interplay of vector insects, plants and phytoplasmas [44].

In order to understand the different symptom manifestation in the three closely related pathosystems, we explored how infections with specific fruit tree phytoplasmas ('*Ca*. P. mali', '*Ca*. P. pyri' and '*Ca*. P. prunorum') of the 16SrX group [11], changed important morphological and physiological parameters of their respective host plants, all Rosaceae [45]. We measured various parameters such as leaf morphology, plant vascular morphology, and callose deposition, determined physical phloem parameters (mass flow velocity and volumetric flow rate, relative density and dynamic viscosity), and analysed the content of several phytohormones in leaf tissues of healthy and phytoplasma-infected plants. The importance of measured parameters for symptom manifestation as well as the impact on vector insects, the epidemiology and phytoplasma spread is discussed.

## Results

### Phytoplasma infection affects leaf and vascular morphology

We first investigated and compared the effects of phytoplasma infection on leaf and vascular morphology. The successful infections resulted in known visible disease symptoms: witches' broom and enlarged stipules in apple trees, premature foliar reddening in pear trees and chlorosis and suberization in peach trees (Fig 1 and Table 1).

The symptoms indicated impairments in the leaf development, which was further examined by a comparison of the leaf lamina, midrib sizes and their ratios among infected and healthy plants. It was found that leaves of AP-infected apple trees were significantly ($p < 0.05$) smaller (length -17% and width -22%) and the diameter of midribs were significantly reduced (-27%) compared to those of healthy plants (Table 2). *Ca*. P. mali infected apple trees had significantly ($p < 0.05$) more leaves than uninfected apple trees (Table 2 and S2 Fig).

The phytoplasma infection in apple trees did not affect the leaf size ratio and the midrib ratio (Fig 1A). In pear, basing upon a significant increase of the leaf width (+8.5%), a significant decrease of the leaf size ratio of nearly 9% was observed, but no changes for the midrib ratio were found (Fig 1B). In contrast to apple and pear plants, phytoplasma infected peach trees exhibited a significant rise of the leaf size ratio of +13% and the midrib ratio of +16% (Fig 1C). No significant changes were found for leaf length, width and midrib diameter (Table 2). All morphological results demonstrated the heterogeneity of the symptoms and indicated differences in the individual host-pathogen interactions.

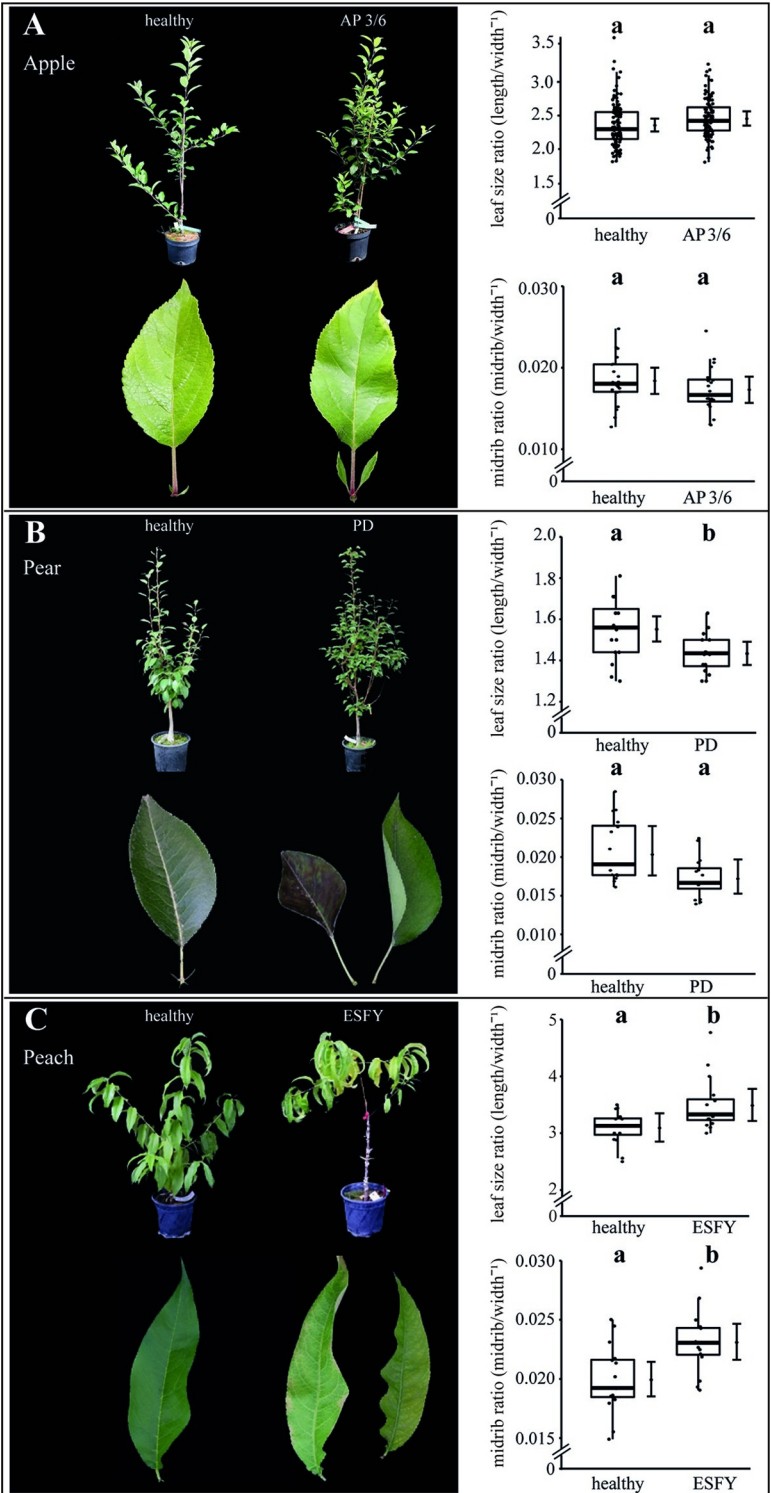

**Fig 1. Symptoms of phytoplasma infected apple, pear and peach trees.** (A) The apple proliferation (AP) induced by '*Candidatus* Phytoplasma mali' led to typical disease symptoms like witches' broom, enlarged and highly serrated stipules. The leaf size ratio (length width[-1]) and the midrib ratio (diameter midrib width[-1]) did not show any effects. (B) The leaves of '*Candidatus* Phytoplasma pyri' affected trees inducing pear decline (PD) were characterized by premature foliar reddening and a significant decrease of the leaf size ratio but not by an impact on the midrib. (C) The leaves of '*Candidatus* Phytoplasma prunorum' infected trees inducing European stone fruit yellows (ESFY) showed chlorosis, and a significant increase of the leaf size and midrib ratio. Boxes represent the interquartile range (IQR) and

whiskers extend to 1.5*IQR. Bars represent the 95% confidence intervals with the estimated marginal means obtained from mixed effect models as dots (both back transformed to the response scale). Letters indicate statistical differences between EMMs of groups at the 0.05 significance level.

The specific impact of the phytoplasma infection on the vascular morphology was investigated by analysing the areas of vascular bundle, xylem, phloem and SEs as well as the ratios of xylem to phloem and SE to phloem (Figs 2–4). For apple, the phytoplasma infection exhibited significantly (p<0.05) reduced sizes of the vascular bundle (-39.1%), xylem (-49.8%), phloem and SE (-33.7%) in comparison to healthy plants, whereas the ratio of SE to phloem was not affected (Fig 2B). Phytoplasma infected pear trees did not show any changes (Fig 3), whereas in peach trees infected with ESFY, the mean sieve element area (-26%) and the ratio of SE to phloem (-46.9%) decreased significantly (Fig 4B). Confirming the heterogeneity of the morphological results (Fig 1), different disease patterns were also found on the cellular level of the vascular system for apple, pear and peach (Figs 2–4).

## The translocation situation and phytohormone distribution are different in the individual host-pathogen-systems

We next examined the consequences of the morphological changes (Figs 2–4) on the physiological situation within the sieve elements. In apple leaves the phloem mass flow velocity and

**Table 1. Overview and summary of the phytoplasma titer determinations (displayed as arithmetic mean of replicates) in apple, pear, and peach.**

|  | plant ID | tissue | phyto- plasma | arithmetic mean of phytoplasma concentration / wet weight [cells g$^{-1}$] | n | plant ID | tissue | phyto- plasma | arithmetic mean of phytoplasma concentration / wet weight [cells g$^{-1}$] | n |
|---|---|---|---|---|---|---|---|---|---|---|
| **Apple** | 2017-16-46 | leaf | - | - | 2 | 2017-18-15 | leaf | AP | 6,37E+08 | 2 |
|  | 2017-17-11 | leaf | - | - | 2 | 2017-18-23 | leaf | AP | 3,53E+08 | 2 |
|  | 2017-17-48 | leaf | - | - | 2 | 2017-18-16 | leaf | AP | 4,62E+08 | 2 |
|  | 2017-17-50 | leaf | - | - | 2 | 2017-18-22 | leaf | AP | 4,96E+08 | 2 |
|  | no No. | leaf | - | - | 2 | 2017-18-4 | leaf | AP | 4,10E+08 | 2 |
|  | 2017-17-2 | leaf | - | - | 2 | 2017-18-49 | leaf | AP | 4,68E+08 | 2 |
|  | 2017-17-9 | leaf | - | - | 2 | 2017-18-1 | leaf | AP | 1,33E+09 | 2 |
|  | 2017-17-19 | leaf | - | - | 2 | 2017-18-25 | leaf | AP | 1,30E+09 | 2 |
|  | 2017-17-36 | leaf | - | - | 2 | 2017-18-34 | leaf | AP | 9,54E+08 | 2 |
|  | 2017-17-39 | leaf | - | - | 2 | 2017-18-27 | leaf | AP | 2,38E+09 | 2 |
|  | 2017-17-42 | leaf | - | - | 2 | 2017-18-46 | leaf | AP | 1,49E+09 | 2 |
|  | 2017-17-5 | leaf | - | - | 2 | 2017-18-7 | leaf | AP | 7,73E+08 | 2 |
|  | 2017-17-17 | leaf | - | - | 2 | 2017-18-21 | leaf | AP | 1,12E+09 | 2 |
|  | 2017-17-7 | leaf | - | - | 2 |  |  |  |  |  |
| **Pear** | 98 | leaf | - | - | 4 | 47 | shoot | PD | 5,66E+07 | 2 |
|  | no No. | leaf | - | - | 4 | 42 | leaf, shoot | PD | 7,82E+09 | 4 |
|  | 2016-8-30 | leaf | - | - | 4 | 55 | leaf, shoot | PD | 6,48E+09 | 2 |
|  | 2016-8-24 | leaf | - | - | 4 | 32 | leaf, shoot | PD | 4,07E+09 | 4 |
| **Peach** | 2017-7-6 | leaf | - | - | 4 | 2017-8-8 | leaf | ESFY | 1,96E+08 | 4 |
|  | 2017-7-1 | leaf | - | - | 4 | 2017-8-6 | leaf | ESFY | 7,77E+08 | 4 |
|  | 2017-7-7 | leaf | - | - | 4 | 2017-8-1 | leaf | ESFY | 2,50E+08 | 4 |
|  | 2017-7-2 | leaf | - | - | 4 | 2017-8-7 | leaf | ESFY | 1,23E+08 | 4 |

AP = apple proliferation, PD = pear decline, ESFY = European stone fruit yellows, n = number of replicates,— = no $C_q$ value or higher than 30 (value of control DNA from healthy trees maintained under insect-proof conditions) are considered as phytoplasma negative.

**Table 2. Mean (±SD) leaf morphology parameters.**

| | Apple | | Pear | | Peach | |
|---|---|---|---|---|---|---|
| parameter | Healthy | AP | Healthy | PD | Healthy | ESFY |
| leaf length [cm] | 11.35 (±1.94)[a] | 9.43 (±1.79)[b] | 6.13 (±0.52)[a] | 6.17 (±0.48)[a] | 12.44 (±1.08)[a] | 11.69 (±1.81)[a] |
| leaf width [cm] | 4.95 (±0.95)[a] | 3.84 (±0.76)[b] | 3.98 (±0.48)[a] | 4.32 (±0.42)[b] | 4.06 (±0.53)[a] | 3.41 (±0.74)[a] |
| diameter midrib [μm] | 916.40 (±124.40)[a] | 667.75 (±161.49)[b] | 821.10 (±141.85)[a] | 740.64 (±69.42)[a] | 799.37 (±95.06)[a] | 782.71 (±173.91)[a] |
| number of leaves per tree | 104 (±17)[a] | 199 (±25)[b] | 257 (±128)[a] | 379 (±117)[a] | 102 (±15)[a] | 76 (±49)[a] |

Different letters indicate significant differences (p < 0.05) between phytoplasma infected and uninfected trees compared within each species. AP = apple proliferation; PD = pear decline; ESFY = European stone fruit yellows. apple: $n_{trees}$ = 14, $n_{leaves}$ = 112; pear: $n_{trees}$ = 10, $n_{leaves}$ = 20; peach: $n_{trees}$ = 8, $n_{leaves}$ = 16.

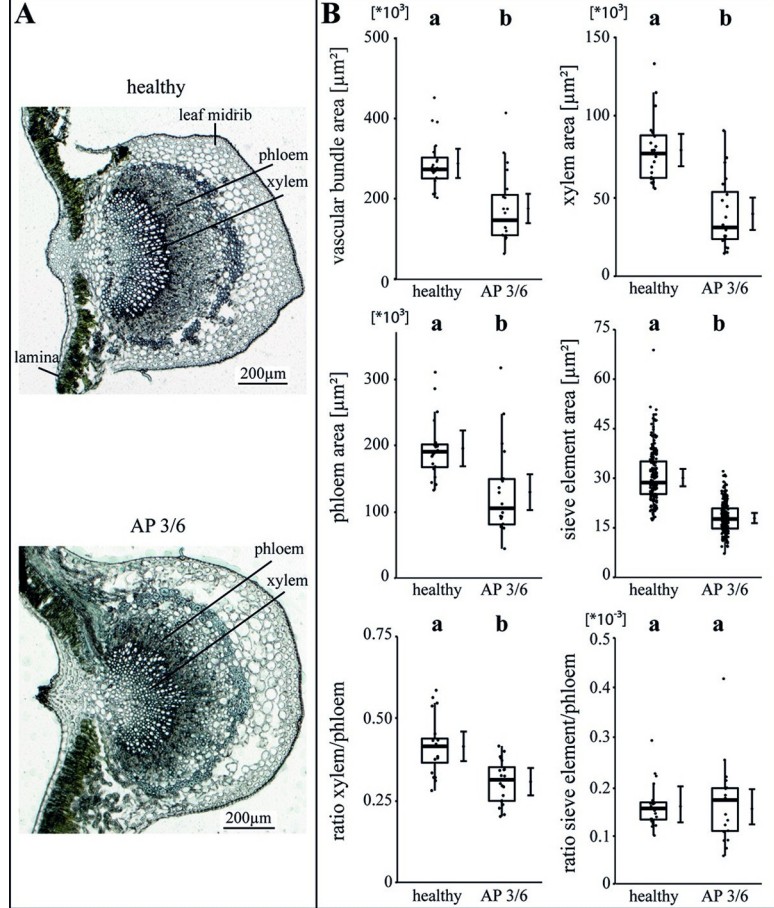

**Fig 2. Analysis of morphological leaf parameters in uninfected and phytoplasma infected apple trees.** (A) The infection of apple trees with a virulent classified 'Candidatus Phytoplasma mali' accession (3/6) was investigated with the morphology of the leaf main vein in the midrib. (B) The morphological analysis consisted of the vascular bundle area, the xylem area, the phloem area, the sieve element area, the xylem/phloem ratio and the sieve element/phloem ratio and showed a significant decrease for nearly all studied parameters in AP infected trees but not for the sieve element/phloem ratio. Box-whisker plots with median as lines and jittered raw values as closed circles (corresponding to each measurement). Boxes represent the interquartile range (IQR) and whiskers extend to 1.5*IQR. Bars represent the 95% confidence intervals with the estimated marginal means obtained from mixed effect models as dots (both back transformed to the response scale). Letters indicate statistical differences between EMMs of groups at the 0.05 significance level.

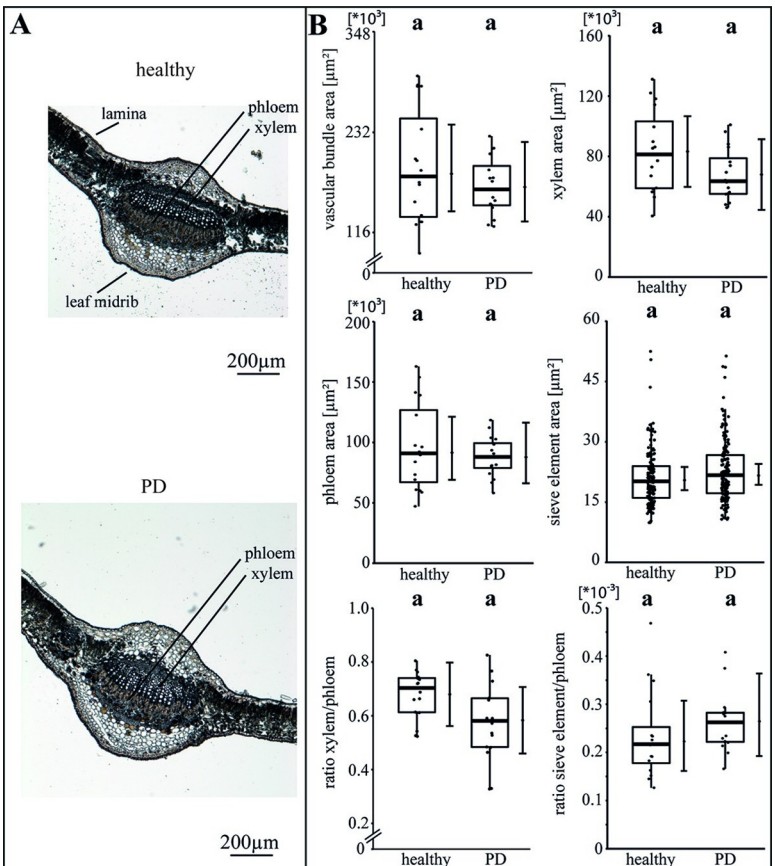

**Fig 3. Analysis of morphological leaf parameters in uninfected and phytoplasma infected pear trees.** (A) The infection of a pear tree with 'Candidatus Phytoplasma pyri' inducing pear decline (PD) was studied by the morphology of the leaf main vein. (B) The morphological analysis consisted of the vascular bundle area, the xylem area, the phloem area, the sieve element area, the xylem/phloem ratio and the sieve element/phloem ratio and showed no significant changes. Box-whisker plots with median as lines and jittered raw values as closed circles (corresponding to each measurement). Boxes represent the interquartile range (IQR) and whiskers extend to 1.5*IQR. Bars represent the 95% confidence intervals with the estimated marginal means obtained from mixed effect models as dots (both back transformed to the response scale). Letters indicate statistical differences between EMMs of groups at the 0.05 significance level.

the calculated volumetric flow rate decreased significantly ($p < 0.05$) in infected leaves in comparison to healthy ones by—25% and -58%, respectively (Fig 5A). In pear leaves the phloem mass flow velocity and the volumetric flow rate increased significantly by +32.6% and +46.6%, respectively (Fig 5B). In peach leaves the phloem mass flow velocity was not affected, but the volumetric flow rate decreased significantly (-30.8%; Fig 5C).

The varying effects for the phloem mass flow indicated changes in the flow properties of the phloem sap. Thus, the dynamic viscosity, density and refractive index of phloem sap obtained by bark tissue centrifugation were measured (Table 3).

No changes of the refractive index for apple and peach were found. In contrast, the dynamic viscosity in infected pear plants was doubled (+104%) and the relative density increased strongly (+97.7%). Unfortunately, the peach plants did not deliver enough phloem sap volume for a complete analysis. For this reason, only the relative density was determined without any significant changes. Furthermore, a comparative analysis of the phloem's relative density between apple, pear and peach revealed significant differences, illustrating a plant specificity of

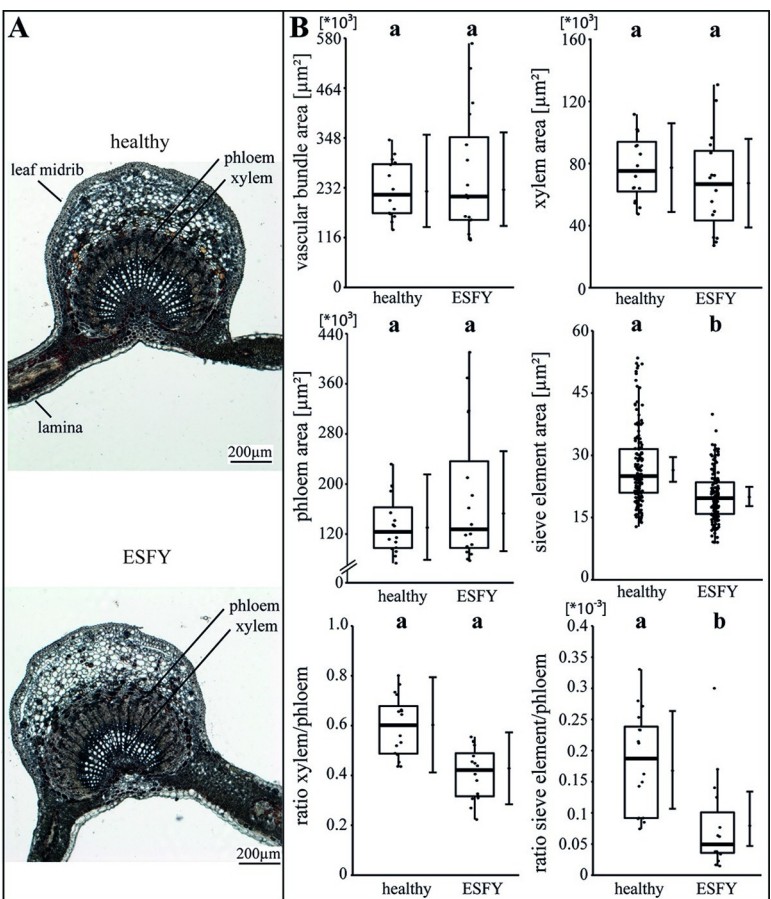

**Fig 4. Analysis of morphological leaf parameters in uninfected and phytoplasma infected peach trees.** (A) The infection of a peach tree with '*Candidatus* Phytoplasma prunorum' inducing European stone fruit yellows (ESFY) was studied by the morphology of the leaf main vein. (B) The morphological analysis consisted of the vascular bundle area, the xylem area, the phloem area, the sieve element area, the xylem/phloem ratio and the sieve element/phloem ratio and showed significant decreases for the sieve element areas and ratios of sieve element to the phloem. Box-whisker plots with median as lines and jittered raw values as closed circles (corresponding to each measurement). Boxes represent the interquartile range (IQR) and whiskers extend to 1.5*IQR. Bars represent the 95% confidence intervals with the estimated marginal means obtained from mixed effect models as dots (both back transformed to the response scale). Letters indicate statistical differences between EMMs of groups at the 0.05 significance level.

the phloem sap composition regarding total sugar content. The measured/calculated phloem mass flow parameters again exhibit heterogeneous effects of a phytoplasma infection (Fig 5 and Table 3) and confirmed the variability of previously shown anatomical/morphological results (Figs 1–4).

To obtain indications for the changed mass flow translocation of the individual plant-phytoplasma variations, the callose deposition in the SEs was visualized and its intensity analysed (Fig 6). No differences of callose depositions were found for the phytoplasma infection, in comparison to healthy apple trees (Fig 6A). In contrast, an increase of callose deposition was found in peach (+300%) and pear (+67%; Fig 6B and 6C), showing a stronger impact on the anatomical and physiological balance in comparison to apple trees.

The stress level of the plants was explored by the measurement of salicylic acid (SA), jasmonic acid-isoleucine (JA-Ile), jasmonic acid (JA), abscisic acid (ABA), 12-oxo-phytodienoic acid (cis-OPDA) and indole-3-acetic acid (IAA) in leaves (Figs 7 and S1 and S7 Table). Upon an infection with the virulent accession 3/6, in apple, SA (+109%), ABA (+55%) and JA-Ile

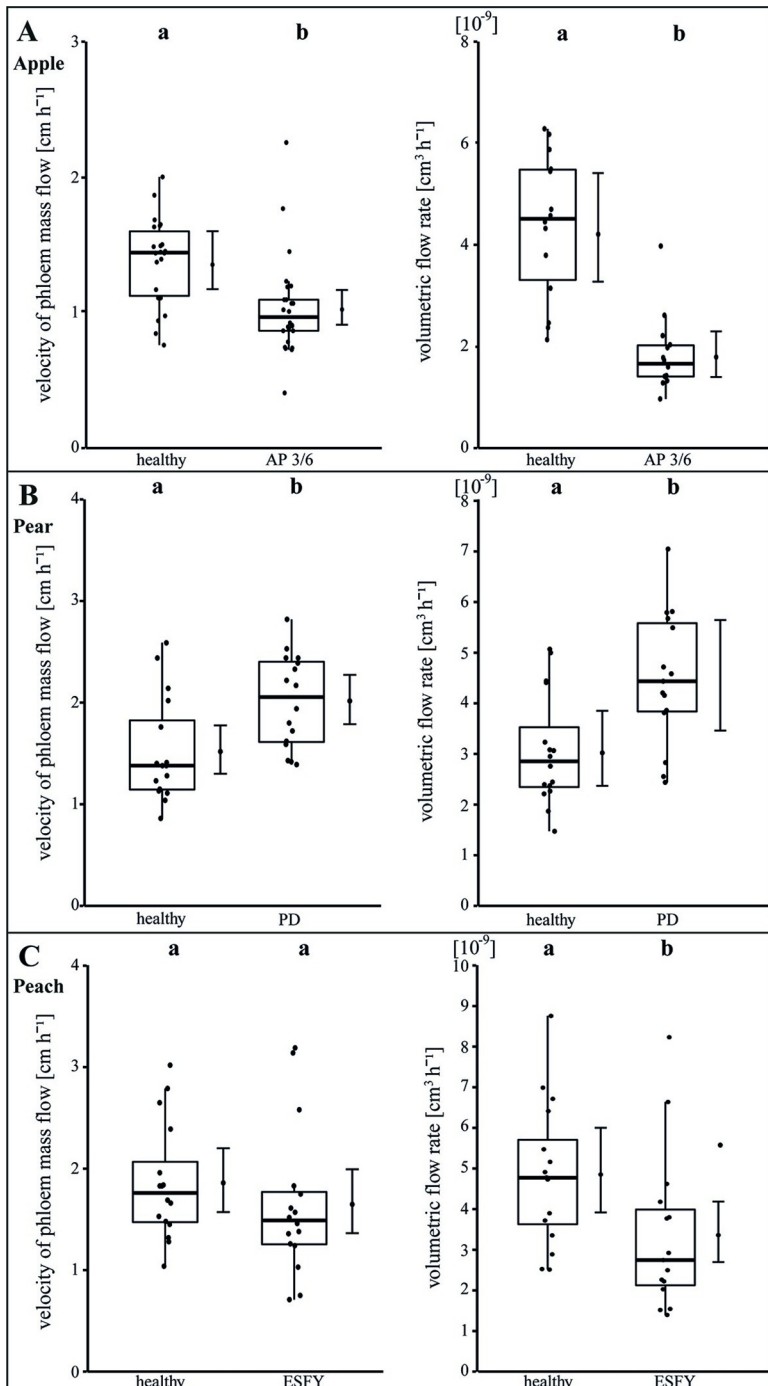

**Fig 5. Analysis of translocation in phloem sieve elements of uninfected and phytoplasma infected apple, pear and peach trees.** The translocation was examined with the determination of the velocity of the phloem mass flow (cm h$^{-1}$) using fluorescence and with the calculation of volumetric flow rates (cm$^3$ h$^{-1}$) in mean single sieve elements. Both parameters were individually determined for (A) apple, (B) pear and (C) peach trees. Apple trees were infected with '*Candidatus* Phytoplasma mali' inducing apple proliferation (AP). Pear trees were infected with '*Candidatus* Phytoplasma pyri' causing pear decline (PD) and peach trees were infected with '*Candidatus* Phytoplasma prunorum' inducing European stone fruit yellows (ESFY). Infected apple trees (AP) showed a significant decrease of phloem mass flow velocity and volumetric flow rates in contrast to infected pear trees (PD) where a significant rise was observed. In infected peach trees (ESFY) the phloem mass flow velocity was not affected but the volumetric flow rate decreased significantly. Box-whisker plots with median as lines and jittered raw values as closed circles (corresponding to each measurement). Boxes represent the interquartile range (IQR) and whiskers extend to 1.5*IQR. Bars represent the 95%

confidence intervals with the estimated marginal means obtained from mixed effect models as dots (both back transformed to the response scale). Letters indicate statistical differences between EMMs of groups at the 0.05 significance level.

(+78%) increased significantly (p<0.05) whereas a significant decrease of cis-OPDA (-45%) and no changes of JA and IAA were observed (Figs 7A and S1). No significant changes for the several measured phytohormones were detected in pear trees due to infection (Fig 7B). In peach trees, SA (+192%), JA-Ile (+345%) and IAA (-40%) were significantly changed in infected plants, whereas JA and ABA did not show any significant changes (Figs 7B and S1). Moreover, the basic level of SA, ABA and cis-OPDA differed among healthy apple, pear and peach plants. For example, ABA was 6-fold higher in pear and 3-fold higher in peach compared to apple.

In accordance with the obtained morphological and functional results, also the effect of phytoplasma infections on the stress-related phytohormone contents revealed different patterns among particular host-pathogen combinations.

## Discussion

Although the phytoplasmas 'Candidatus Phytoplasma mali´, 'Ca. P. prunorum´ and 'Ca. P. pyri´ belong to the same 16SrX group, their pathogenicity is quite different in their respective host plants Malus domestica (apple), Prunus persica (peach) and Pyrus communis (pear). Apple trees can survive a phytoplasma infection for decades, whereas phytoplasma-infected peach and pear trees often die after a few years and sometimes even after a few weeks (quick decline of pear) [3,4,9,46,47]. This indicates a higher tolerance resulting in better survival rate for phytoplasma-infected apple compared with pear and peach; however, the underlying reason is not known yet. To address this open question, we collected a comprehensive dataset (summarized in Table 4) covering anatomical and physiological responses of each plant species to its particular phytoplasma infection, supporting a co-evolutionary impact. Based on these results we are discussing several implications thereof in the following paragraphs.

### Phytoplasma infections affect the vascular morphology of apple trees more than peach and pear trees

Although apple trees survive phytoplasma infection for decades [47], various significant reductions in size were found in leaves (width, length, midrib), tissues (vascular bundle, phloem and xylem) and cells (sieve elements) when compared to non-infected plants (Figs 1 and 2). In contrast, pear and peach trees showed less significant differences between healthy and phytoplasma-infected leaves; if any, we found significant increases for leaf size and midrib ratio for

**Table 3. Mean (±SD) functional/physiological parameters.**

| parameter | Apple | | Pear | | Peach | |
|---|---|---|---|---|---|---|
| | Healthy | AP | Healthy | PD | Healthy | ESFY |
| refractive index [˚Brix] | 9.643 (±1.200)[a] | 9.423 (±1.018)[a] | 7.250 (±1.909)[a] | 14.333 (±6.280)[b] | 11.727 (±2.494)[a] | 10.714 (±2.079)[a] |
| dynamic viscosity [mPa s] | 4.078 (±0.652)[a] | 3.916 (±1.093)[a] | 1.957 (±0.368)[a] | 3.999 (±2.035)[b] | n.d. | n.d. |
| density [g L$^{-1}$] | 996.5 (±94.9) [a] | 1003.6 (±133.4) [a] | 1001.4 (±50.8)[a] | 1058.1 (±55.7)[a] | n.d. | n.d. |

Various physicochemical parameters (refractive index, dynamic viscosity and density) were determined for the centrifugates of the apple, pear and peach bark. Different letters indicate significant differences (p < 0.05) between phytoplasma infected and uninfected trees compared within each species. AP = apple proliferation; PD = pear decline; ESFY = European stone fruit yellows; mPa = milli Pascal; s = second; g = gram; L = litre.

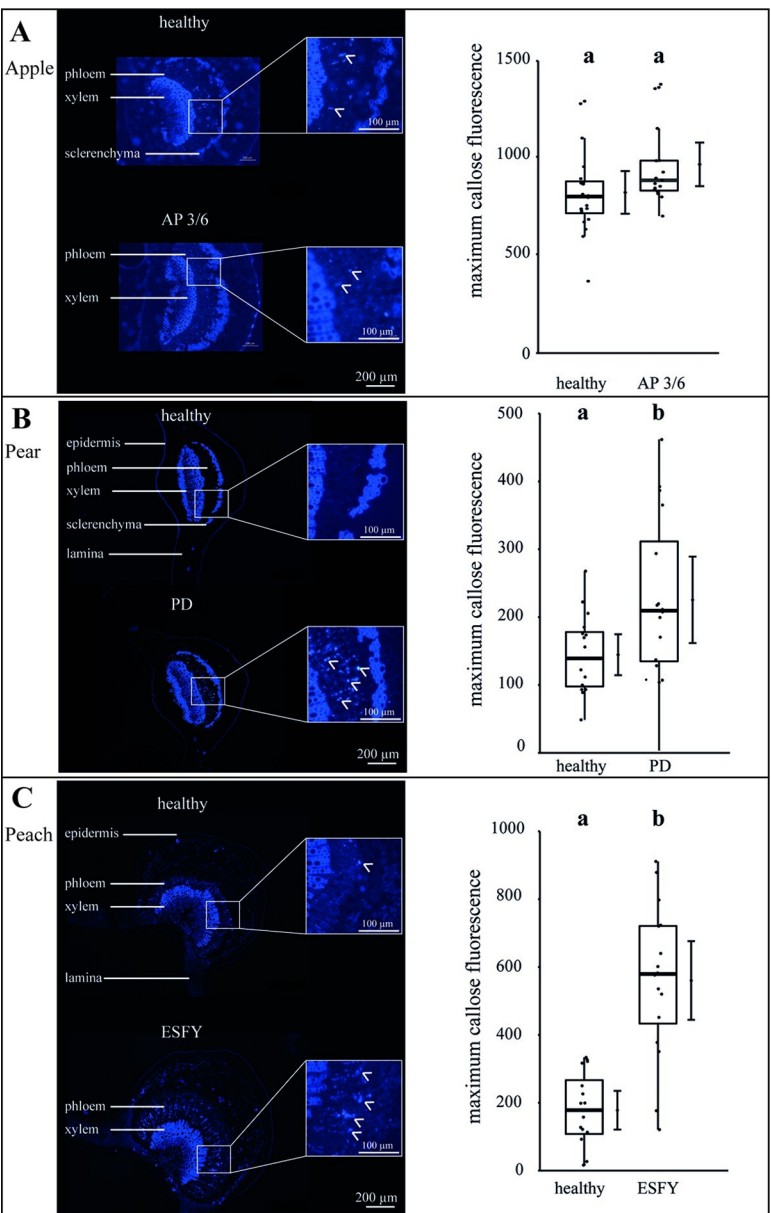

**Fig 6. Analysis of callose deposition in the leaf phloem tissue of uninfected and phytoplasma infected apple, pear and peach trees.** At cross sections of the leaf mid rip, the callose deposition in sieve elements was stained with aniline blue and detected via fluorescence microscopy (see panels on the left side). The callose fluorescence was quantified after subtracting auto-fluorescence (see panels on the right side). (A) In apple trees, an infection with the virulent accession (3/6) inducing apple proliferation (AP) did not show any differences in the callose deposition in comparison to uninfected plants. (B+C) The phytoplasma infection of pear trees (PD) and peach trees (ESFY) induced a significant (p<0.05) increase of callose deposition in sieve elements. Box-whisker plots with median as lines and jittered raw values as closed circles (corresponding to each measurement). Boxes represent the interquartile range (IQR) and whiskers extend to 1.5*IQR. Bars represent the 95% confidence intervals with the estimated marginal means obtained from generalized least square models as dots (both back transformed to the response scale). Letters indicate statistical differences between EMMs of groups at the 0.05 significance level.

peach and leaf width for pear (Table 4). That seems surprising as it could be expected that plants with a higher tolerance and survival rate would show a lower rate of symptoms than plants demonstrating a higher mortality [4]. Reduced leaf sizes in apple trees results directly in a drop of photosynthesis that may be compensated by the higher number of leaves shown with

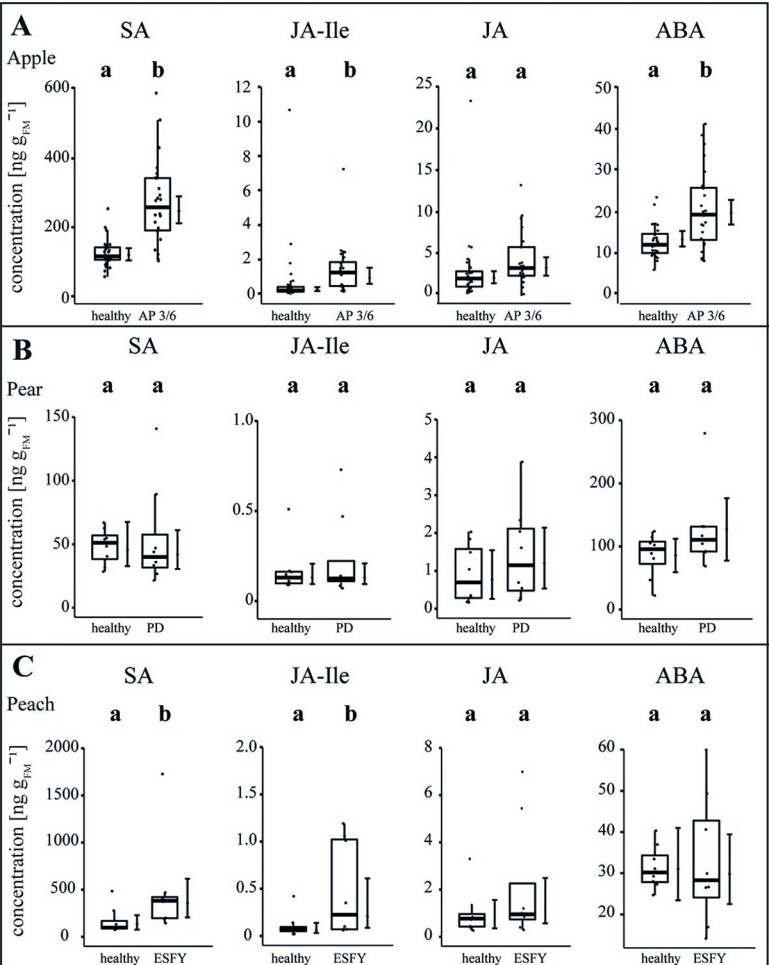

**Fig 7. The phytohormone balance in uninfected and phytoplasma infected apple, pear and peach trees.** The concentrations (ng $g_{FM}^{-1}$) of several phytohormones–salicylic acid (SA), jasmonic acid-iso leucine (JA-Ile), jasmonic acid (JA) and abscisic acid (ABA)–were measured in the leaves of uninfected and phytoplasma infected (A) apple, (B) pear and (C) peach. For apple, a virulent accession (3/6) was considered inducing apple proliferation (AP). Pear trees showed pear decline (PD) and peach trees showed the European stone fruit yellows (ESFY). Box-whisker plots with median as lines and jittered raw values as closed circles (corresponding to each measurement). Boxes represent the interquartile range (IQR) and whiskers extend to 1.5*IQR. Bars represent the 95% confidence intervals with the estimated marginal means obtained from linear models as dots (both back transformed to the response scale). Letters indicate statistical differences between EMMs of groups at the 0.05 significance level.

the bushy habitus (Fig 1). The increased number of leaves may produce and release more phytoplasma induced VOCs (volatile organic compounds) to attract vector insects, which detect appropriate host plants for feeding and reproduction [37,38,41–43]. By analysing VOCs emitted by the leaves of apple trees, it was shown that '*Ca*. P. mali' changed the VOCs composition of infected trees compared to healthy ones by inducing the production of the sesquiterpene β-caryophyllene [39,40]. *Cacopsylla picta* emigrants are attracted by β-caryophyllene and lured to infected apple trees [40], before migrating to their overwintering host. This behaviour increases the number of psyllids, which are able to acquire '*Ca*. P. mali' and enhance the spread of the phytoplasma diseases [48].

The morphological and physiological changes in apple trees might enable plants to handle a phytoplasma infection as a result of a selective adaptation. Intraspecific comparisons of these

**Table 4. Comparison of all morphological and physiological results of phytoplasma infected apple, pear and peach trees.**

| | Apple | Pear | Peach |
|---|---|---|---|
| number of leaves per tree | + | = | = |
| leaf size ratio [length/width] | = | - | + |
| midrib ratio [diameter midrib/leaf width] | = | = | + |
| diameter midrib [μm] | - | = | = |
| leaf length [cm] | - | = | = |
| leaf width [cm] | - | + | = |
| vascular bundle [μm$^2$] | - | = | = |
| xylem [μm$^2$] | - | = | = |
| phloem [μm$^2$] | - | = | = |
| sieve elements [μm$^2$] | - | = | - |
| ratio xylem/phloem | - | = | = |
| ratio sieve element/phloem | = | = | - |
| mass flow [cm h$^{-1}$] | - | + | = |
| volumetric flow rate [cm$^3$ h$^{-1}$] | - | + | - |
| dynamic viscosity [mPa s] | = | + | n.d. |
| refractive index [°Brix] | = | + | = |
| absolute density [g L$^{-1}$] | = | = | n.d. |
| callose intensity | = | + | + |
| salicylic acid [ng g$_{FM}^{-1}$] | + | = | + |
| jasmonic acid-iso leucine [ng g$_{FM}^{-1}$] | + | = | + |
| jasmonic acid [ng g$_{FM}^{-1}$] | = | = | = |
| abscisic acid [ng g$_{FM}^{-1}$] | + | = | = |
| 12-oxo-phytodienoic acid [ng g$_{FM}^{-1}$] | - | = | = |
| Indole-3-acetic acid [ng g$_{FM}^{-1}$] | = | = | - |

All studied parameters are listed in the first column and significant changes in comparison to healthy plants are shown for the considered plant species–increase (+), decrease (-), no change (=) and not determined (n.d.).

parameters between plant genotypes of different sensitiveness to the same phytoplasma isolate should be used in future studies to confirm or refuse this evolution-based hypothesis.

Nii (1993) described differences in ingrowth structures in the SEs of the phloem among single Rosacea species and found rather similar structures in *Malus* and *Pyrus* in comparison to *Prunus* [49]. In theory, the ingrowth structures might influence the phloem mass flow due to reduced size and/or volume of SEs and thus, reflect a higher resistance for the phloem mass flow. Beside this, phytoplasma colonization of the SEs might affect the phloem mass flow (= plant physiology) negatively. Liu and Gao (1993) found similar phloem structures in *Malus* and *Pyrus* in comparison to *Prunus*, too [50]. The SEs in *Prunus* were shorter and each SE was connected to 2 to 3 companion cells (CCs) in comparison to *Malus* and *Pyrus*. In theory, the *Prunus* SEs have a higher flow resistance because of their shortness and increasing number of sieve plates, which might be balanced with a higher amount of CCs per SEs. The CCs guarantee the physiological function of the SEs and a rise of CCs per SEs might enable the *Prunus* species to establish higher pressures, higher viability or turnover. In particular, the *P. communis*/ '*Ca*. P. pyri´–system is of high scientific interest, as disease severity shows huge variance from mild symptoms, such as premature foliar reddening, to severe growth depression and the quick decline of infected trees [9].

## The physiological phloem parameters in pear and peach are more affected than in apple

All observed results regarding the particular morphological (Figs 1–4) and functional measurements (Fig 5) illustrate well the consequences of a phytoplasma infection for a plant. However, depending on the individual host-pathogen system, they are heterogeneous between the systems and specific within. One reason for the heterogeneity might be found in the formation of plant defence. A general defence response to several (a)biotic stresses is an elevated $Ca^{2+}$-dependent deposition of callose that was already reported for phytoplasma infections [51,52]. We were able to show that *P. communis* and *P. persica* trees responded to phytoplasma infections by blocking sieve plates with callose. Phytoplasma effectors may cause regulation of $Ca^{2+}$ channels, which leads to sieve-tube occlusion with dramatic effects on photoassimilate distribution as indicated by the reduced volumetric flow rate in *P. persica* trees. Surprisingly, the mass flow of infected *P. communis* trees was increased though a simultaneous increase of phloem sap viscosity, which reflects an increased sugar content. The reason may be an increased pressure gradient (we calculated ~6.5 bar using the van-'t-Hoff equation) in infected trees, which drives the mass flow against the resistance of the SEs. Thus, *P. communis* trees have to take major effort with increased energy supply, which eventually causes negative feedback. As the callose deposition in response to phytoplasma infections never results in a restriction of the bacteria [53], it suggests that callose deposition is a costly non-functional leftover of general defence mechanisms only and for instance also found in answer to a '*Candidatus* Liberibacter asiaticus' infection [54]. Strikingly, the infection with '*Ca*. P. mali' in apple trees did not lead to an increased callose deposition. This might be due to the particular apple cultivar, phytoplasma strain, unknown defence mechanisms or an evolutionary adaptation to the phytoplasma infection. Whether the callose deposition is directly or indirectly induced by phytoplasmas, is an issue for future studies.

Callose deposition also is a defence mechanism against phloem-feeding and is induced by phloem feeding insects [55,56]. Therefore, callose concentrations are of great importance for phloem-feeding vector insects carrying phytoplasmas. The occlusion of sieve tubes inhibits the phloem flow and affects the feeding of piercing-sucking insects on the phloem tissue of host plants [57]. Nevertheless, the brown planthopper *Nilaparvata lugens* is able to overcome this plant defence by activating and secreting a hydrolysing enzyme, which induces the degradation of callose in SEs [55]. Whether psyllid species transmitting AP, PD and ESFY have evolved such mechanisms to overcome this particular plant defence is still unknown. Yet, it was shown that phloem ingestion of *C. pruni* was not influenced by phytoplasma infection of its host plants (*P. persica* and *P. insititia*) [32]. The increased callose deposition did not affect all SEs as we still measured mass flow in SEs (Table 4), meaning that plum psyllids could feed on callose-free SEs.

In general, sugars (e.g. sucrose) are known to stimulate feeding of phloem-feeding insects such as aphids [58,59]. Thus, the detected higher sugar concentration in infected pear phloem (Table 4) might increase probing and feeding behaviour of pear psyllids and, therefore, increase the acquisition and spread of phytoplasmas in pear orchards. However, a recent detailed phloem composition analysis of *Prunus* trees revealed no major differences in the phloem metabolite composition between ESFY infected and healthy trees [32]. In contrast, the soluble sugar content increased and the composition of phloem metabolites differed significantly between *Ca*. P. mali infected and non-infected apple plants [35]. Furthermore, sugar and sugar alcohol levels increased in diseased plants, while organic and amino acid content remained constant. A broad influence of the carbohydrate situations and changed source-sink relations in plants were already described following a '*Candidatus* Liberibacter' or Spiroplasma infection [60,61].

## Phytohormones are affected in apple and peach but not in pear

Besides the direct and local defence mechanisms, an activation and systemic distribution of signalling compounds such as phytohormones via the SEs, may be induced by phytoplasmas and may have an impact on the plant's defence. Moreover, typical symptoms, such as development of witches' brooms, smaller fruits, reduced leaf and vascular morphology of diseased apple trees, can be explained with an infection-induced imbalance of phytohormones [25]. In apple and peach trees, SA and JA-Ile levels significantly increased in infected trees, indicating the involvement of defence pathways to phytoplasma colonization. Furthermore, the content of ABA in apple leaves increased as well. Commonly, SA plays the central role for the interaction between biotrophic pathogens and host plants [62,63] and an increase of SA in apple trees after *Ca.* P. mali infection was found earlier [25]. In contrast, the JA pathway is induced in response to wounding, herbivore attack and necrotrophic pathogens [64]. The development of different pathways in reaction to different threads enables plants to respond more specifically and is therefore more resource-efficient. An antagonistic crosstalk between JA/ABA and SA was detected in several plant species [25,65]. Not surprisingly, some bacteria species utilize SA hydroxylase for a degradation of SA to catechol to suppress an adequate plant defence reaction [66] or evolved the production of effector proteins that interfere with SA regulated defence responses by activating JA pathway [67]. This mechanism was also detected for phytoplasmas in Aster yellows-witches' broom phytoplasma (AY-WB). AY-WB produces the SAP11 effector that down-regulates the plant defence response by reducing lipoxygenase2 expression and JA production [68]. This down-regulation of defence mechanisms in AY-WB infected plants is advantageous to vector fitness [68]. Recently, an SAP11-like protein was detected in '*Ca.* P. mali' that affected JA, SA and ABA pathways [69,70]. JA plays a central role in induced plant defence e.g. by regulating the biosynthesis of herbivore-induced plant volatiles [64]. Moreover, exogenous application of JA can be used to elicit plant defence responses similar to those induced by biting-chewing herbivores and mites that pierce cells and consume their contents. A low-dose JA application results in a synergistic effect on gene transcription and an increased emission of a volatile compounds involved in indirect defence after herbivore infestation [71]. The induction of JA defence mechanisms in apple, pear and peach in response to psyllid feeding has not yet been proven. However, infestations of *Citrus* plants with Asian citrus psyllid (*Diaphorina citri*) led to an upregulation of genes involved in the JA-pathway [72]. Additionally, the infection of Citrus trees with the phloem dwelling proteobacterium *Candidatus* Liberibacter asiaticus induced the SA-pathway [72] and resulted in an increased emission of methyl salicylate from infected plants [73]. Auxins (IAA and IBA) were shown to induce the recovery of periwinkle plants from '*Ca.* P. pruni' and '*Ca.* P. asteris' infections [74], illustrating the importance of IAA in plant-pathogen interactions. Interestingly, the IAA concentration in infected *P. persica* plants was significantly lowered compared to healthy peach trees. A reduced auxin content was also detected in leaves of lime infected with '*Ca.* P. aurantifoliae' [75]. Imbalanced auxin concentrations might be responsible for abnormal growth of infected peach trees (Fig 1C).

## What are possible consequences for the plant population?

The detected morphological changes and physiological reactions towards a phytoplasma infection in apple trees are beneficial for the individual apple plant, enabling it to survive with the pathogen, but may have negative effects on the population level. The survival of infected trees provokes the accumulation of the pathogen in the plant population and increases the chances of new infections of vector insects and the spread of the pathogen. Formation of dwarfed fruits is a common symptom of *Ca.* P. mali infected apple trees, but might be compensated by heavy

bearing of trees [76]. A negative impact on plant population might have emerged by reduced reproduction of infected individuals. In contrast, an infection with *Ca*. P. pyri causes the death of pear trees and may limit pear to pear infections. Infected potted *P. persica* trees showed a more reduced lifespan in our experiments than healthy plants and died within 2–3 years after infection. Different *Prunus* species showed important differences in the susceptibility towards both phytoplasma and vector [36,48]. Wild *Prunus* species cope better with the phytoplasma infection than *P. persica* cultivars and represent an infection reservoir [48]. Investigations of the impact of phytoplasmas on reproductive organs, seed numbers, seed viability and fitness of progeny are required to evaluate the impact of the phytoplasma infection on the population development. The reproductive fitness of cultivars is not relevant in fruit farming, as the common practice is to propagate cultivars by grafting. Thus, influence of phytoplasmas on plant reproduction is only relevant for the wild ancestors of cultivated fruit crops, while our investigations have been done in non-natural farming systems with cultivated fruit trees. Additionally, it remains unclear, whether seed transmission of these phytoplasmas is possible or not. Future studies elucidating these aspects are necessary.

## Conclusion

In the three investigated fruit crops the infection with specific phytoplasmas induced different morphological and physiological responses in the particular host plants. As apple trees generally survive a phytoplasma infection more often and much longer than peach and pear, some unique apple-specific responses are most interesting and indicative features that could explain how a plant might become tolerant against phytoplasma. Based on the results obtained, the long-lasting changes in the structure of the vascular system with all physiological consequences on the sap flow found in apple trees provides a promising step towards a deeper understanding of host plant defence against phytoplasmas. Despite the growing understanding of this patho-system, it seems clear that the complexity of these interactions is not fully elucidated, yet, and many open questions remain: Does the plant perceive a phytoplasma infection at all? If so, what does the plant recognize? Is there a MAMP/DAMP/effector present that induces an increased defence response in the SEs? What are the specific events during infections in the host on spatio-temporal and intensity level? How do the antagonists interact on the molecular level? All these questions require more investigations on the molecular level including for example RNAseq and transgenic approaches.

## Materials and methods

### Plant material and phytoplasma inoculation

Apple trees (*Malus domestica*) cv. 'Gala Royal' were grown on clonal rootstock cv. 'M9' (non-infected control, n = 14), pear trees (*Pyrus communis* L.) cv. 'Williams Christ' were grown on the *P. communis* rootstocks cv. 'Kirchensaller Mostbirne' (non-infected control, n = 5), and peach trees (*Prunus persica* (L.) cv. 'South Haven' were grown on peach seedlings cv. 'Montclar' (non-infected control, n = 4). Additional apple, pear and peach plants were inoculated by grafting of two buds from trees infected with the respective phytoplasma. Apple trees were infected with a virulent accession (3/6, n = 13) in 2017 [77–79]. Pear trees were infected with 'Ca*. P. pyri' (PD-W, n = 5) in 2012 (Table 5). This strain causes mild symptoms but no quick decline [9], peach trees were infected with 'Ca*. P. prunorum' (ESFY-Q06, n = 4) in 2017 (Table 5). Trees used as phytoplasma sources were cultivated in insect save environments. Experiments with pear and peach trees were conducted in 2018. Apple trees were investigated in 2019. All plants were grown under natural conditions in an insect safe environment.

**Table 5. Biological and geographical origin of phytoplasma strains.**

| Phytoplasma isolate | Geographic Origin | Additional Information | Description |
|---|---|---|---|
| PD-W | United Kingdom, from David L. Davies | Source: maintained in *in-vitro* culture at the Al Planta—Institute for Plant Research, RLP AgroScience [80] | [81] |
| ESFY-Q06 | Neustadt a. d. Weinstraße, Germany | Source: 'Ca. P. prunorum' infected *C. pruni* individuals caught in the course of transmission studies carried out 2006 at the experimental field of the AlPlanta-IPR, RLP AgroScience, Neustadt a.d. Weinstraße, Germany by Wolfgang Jarausch. | this study |
| AP 3/6 | Dossenheim, Germany | Virulence: severe | [77–79] |

## DNA extraction of phytoplasmas

DNA from leaves and phloem scrapings was isolated using a cetyltrimethylammonium bromide extraction method. The applied method for extraction of nucleic acids from woody plants with modifications from Appendix 1 of EPPO diagnostic protocols for the detection of phytoplasmas [82,83] is based on Doyle and Doyle [84]. Due to irregular distribution of 'Ca. P. pyri' in the apex [85], in part infection status of pear trees was confirmed by extraction of phloem scrapings from shoots of PD inoculated trees, in addition to extraction of leaf tissue from mass flow measurements. Leaves and phloem scrapings were ground in preheated extraction buffer (60°C, 2.5% (w/v) cetyltrimethylammonium bromide, 1.4 M NaCl, 20 mM EDTA, 100 mM Tris-HCl pH 8.0, 1% (w/v) polyvinylpyrrolidone 40, 0.2% (v/v) 2-mercaptoethanol (with a tissue/buffer ratio 1:10; 0.1 g of tissue in 1 ml buffer) using a homogenizer (BIOREBA AG, Reinach, Switzerland) in extraction bags (BIOREBA AG). Homogenate (1 ml) was transferred into a microcentrifuge tube and incubated at 60°C for 30 min. An equal volume of chloroform was added, the tube was briefly vortexed and shaken for 5 min at room temperature. After a centrifugation step (10,000 *g*, 6 min at room temperature, Heraeus Fresco 17 Microcentrifuge, Thermo Fisher Scientific, Dreieich, Germany) the aqueous phase was transferred into a new centrifuge tube. For precipitation of nucleic acids an equal volume of isopropanol was added, the tube was inverted and incubated at 4°C overnight. Precipitate was recovered by centrifugation at 10,000 *g* for 10 min at room temperature. Supernatant was discarded and the nucleic acid pellet was washed with 70% ethanol (centrifugation step at room temperature, 10,000 *g*, 10 min), air dried and resuspended in 50 µl high-performance liquid chromatography (HPLC) water (VWR International GmbH, Bruchsal, Germany). Unless explicitly stated elsewhere, laboratory chemicals were purchased from Carl Roth GmbH (Karlsruhe), Bernd Kraft GmbH (Duisburg) and Sigma-Aldrich Chemie GmbH (Taufkirchen), Germany, respectively.

## Real-time PCR

Quantitative PCR (qPCR) was performed with the Bio-Rad CFX96 Thermal Cycler (Bio-Rad Laboratories GmbH, Munich, Germany) using primer pair and probe of a TaqMan assay developed by Christensen *et al.* [86] for the generic detection of phytoplasma DNA. The amplification of a part of the 16S rDNA gene was performed in 25 µl reactions containing 1 µl of DNA extraction, 0.625 U of FastGene Taq DNA Polymerase (Nippon Genetics Europe GmbH, Düren, Germany) with provided 10 x reaction buffer A (with 1.5 mM MgCl$_2$), 0.5 µl of dNTPs (10 mM each, Steinbrenner Laborsysteme GmbH, Wiesenbach, Germany), 1 µl of each primer (10 µM, Eurofins Genomics Germany GmbH, Ebersberg, Germany), 0.5 µl of TaqMan probe (10 µM, Eurofins Genomics Germany GmbH), and HPLC water (VWR International GmbH). Amplification parameters were 15 min at 95°C followed by 46 cycles at 95°C for 15 s and 60°C for 1 min. Data analysis was performed with the BioRad CFX Manager 3.0 software (Bio-Rad Laboratories GmbH, Munich, Germany).

## Calculation of phytoplasma titre

Phytoplasma concentration (number of copies of phytoplasma 16S rDNA gene per μl) was calculated automatically from the quantification cycle ($C_q$) values by the use of a cloned 16S rDNA gene standard ranging from $10^1$ to $10^9$ copies in qPCR with the internal manufacturer's software (referred to as estimated qPCR concentration). Samples with $C_q$ values higher than 30 were considered as tested negative (see Table 1; negativity threshold was determined by the use of the cloned 16S rDNA gene standard and negative controls (DNA from healthy trees maintained under insect-proof conditions)). Calculation of phytoplasma cells per gram wet weight of extracted leaf tissue and phloem scrapings was performed by multiplying the assessed qPCR concentration by the applied volume of extraction buffer (Y) and the volume of HPLC water used for DNA resuspension, by dividing by the number of 16S rDNA operons (2) and the wet weight of extracted leaf tissue (X):

$$Phytoplasma\ titer \left[\frac{cells}{g}\right] = qPCR\ concentration \left[\frac{copies}{\mu l}\right] * \left( Y * 50 * \frac{1}{2} * \frac{1}{X} \right) \left[\frac{\mu l}{g}\right] \qquad (1)$$

## Experimental set-ups

From each tree, two (apple) or four (pear and peach) mature leaves with similar symptoms were randomly selected. The mass flow measurements were done *in vivo*. Thereafter, the leaves were cropped, maximum length and width of the leaf lamina were measured. Additionally, length and width of six leaves were measured from apple trees. Cross sections were done from leaves used for mass flow measurements to analyse the plant vascular morphology and callose deposition. The middle part of the midrib was fixed in a fixative (see below), the rest of each leaf was used for phytoplasma titre determination. Leaves for phytohormone measurements were separately collected, immediately frozen in liquid nitrogen and stored at -20˚C until used.

## Determination of symptoms and leaf morphology

Next to the general observation and assessment of the known symptoms following a phytoplasma infection the impact on the morphology of apple, pear and peach leaves was investigated with the determination of the length and maximum width of the leaf lamina. Photographs (Canon EOS 760D, Canon Deutschland GmbH, Krefeld, Germany) were taken to document characteristic symptoms at whole plant and leaf level.

## Microscopic analyses of the plant vascular morphology and callose deposition

Cross sections of the midribs were created halfway from the base to the tip of each leaf. Therefore, pieces of about 1 x 1 cm were fixed in 2.5% (w/v) glutaraldehyde, 2% (v/v) paraformaldehyde in 0.1 M sodium-potassium phosphate buffer (pH 7.4, Merck KGaA, Darmstadt, Germany). Sections were cut at a thickness of 20 μm with a cryostat (Leica JUNG CM3000, Leica Microsystems, Wetzlar, Germany) at a chamber temperature of -26˚C and a specimen head temperature of -23˚C. Pieces were bound to a specimen disc by embedding them into plant tissue freezing medium (Jung, Leica Microsystems, Wetzlar, Germany) and then frozen at the quick freeze shelf for 10 min prior sectioning. Each cross section was stained for at least 30 min with 0.1% aniline blue solution (Sigma Aldrich, St. Louis, Missouri, USA) to visualize callose deposition at sieve plates.

Each cross section was imaged using an AXIO Imager.M2 (Zeiss Microscopy GmbH, Jena, Germany) equipped with a 10x objective (N-Achroplan 10x/0.3) and a 40x objective (W

N-Achroplan 40x/0.75). The bright field and fluorescence images were recorded with a colour camera (AXIOCAM 503 colour Zeiss, Jena, Germany) by use of a DAPI (EM 445/50 nm) filter. Each digital image from infected and uninfected trees was analysed with the determination of (1) the diameter of midribs, (2) the area of the vascular bundle, (3) the xylem area, (4) the phloem area and (5) the area of 10 sieve elements per section using the ZEN software (Zeiss, Jena, Germany). The digital images were processed with ZEN software and edited with Adobe PhotoShop to optimize brightness, contrast and colouring. The intensity of aniline blue fluorescence was measured using ZEN software by analysing the whole phloem area as region of interest (ROI) and ROIs of healthy and infected plants were comparatively evaluated.

## Determination of the phloem mass flow velocity

The phloem mass flow rate was measured with the phloem mobile fluorochrome 5,6-carboxyfluorescein diacetate (CFDA) dye (ThermoFisher Scientific, Waltham, Massachusetts, USA). CFDA permeates the plasma membrane in the non-fluorescent acetate form and is cleaved by cytosolic esterases generating membrane-impermeant fluorescent carboxyfluorescein (CF) (handbook from Molecular Probes, Eugene, OR, USA). CF is trapped inside SEs and transported by mass flow in the sieve tubes. A stock solution was prepared by solubilisation of 1 mg CFDA in 1 ml DMSO. A working solution of 1 μl stock solution in 1 ml buffer solution (containing 2 mol m$^{-3}$ KCl, 1 mol m$^{-3}$ CaCl$_2$, 1 mol m$^{-3}$ MgCl$_2$, 50 mol m$^{-3}$ mannitol, and 2.5 mol m$^{-3}$ MES/NaOH buffer, pH 5.7) was applied at a cut leaf tip. After an inoculation period of 1 to 2 h at room temperature, each leaf was removed from the plant. Immediately, cross sections of the mid ribs were made by hand with a sharp and fresh razor blade in one-centimetre intervals from the basal side of the leaf. Sections were covered with distilled H$_2$O, a cover glass and examined for appearance of fluorescence emitted from CF (emission 510–580 nm), by the means of an inverted fluorescence microscope (AxioVert S100, Carl Zeiss, Jena, Germany). The transport velocity was calculated by dividing the measured distance the CF moved within the sieve elements from the application side towards the leaf base by the exact inoculation time (from dipping one leaf tip into CFDA to removing of the specific leaf from the plant).

## Calculation of the volumetric flow rate

The volumetric flow rate ($J_V$) was calculated by multiplying the measured phloem mass flow velocity ($V_a$) by the median area of ten measured sieve elements ($\tilde{A}_{SE}$):

$$J_v\left[\frac{cm^3}{h}\right] = V_a\left[\frac{cm}{h}\right] * \tilde{A}_{SE}[cm^2] \tag{2}$$

## Determination of phytohormones

Four leaves were harvested of each tree, immediately frozen in liquid nitrogen, and stored at -20°C. The leaves of each tree were pooled and 250 mg (per sample and two samples for each tree) were homogenized using a Geno/Grinder (Spex SamplePrep, Stanmore, UK) at 1100 rpm for 1 min and extracted in 1.5 ml methanol containing 60 ng D4-SA (Santa Cruz Biotechnology, USA), 60 ng D6-JA (HPC Standards GmbH, Germany), 60 ng D6-ABA (Santa Cruz Biotechnology, USA), 12 ng D6-JA-Ile (HPC Standards GmbH), and D5-indole-3-acetic acid (D5-IAA, OlChemIm s.r.o., Olomouc, Czech Republic) as internal standards. Samples were agitated on a horizontal shaker at room temperature for 10 min. The homogenate was mixed for 30 min and centrifuged at 13,000 rpm for 20 min at 4°C; the supernatant was collected. The homogenate was re-extracted with 500 μl methanol, mixed and centrifuged; the supernatants were pooled. The combined extracts were evaporated under reduced pressure at 30°C and dissolved in 500 μl methanol.

Phytohormone analysis was performed by LC-MS/MS as in Heyer et al. [87] on an Agilent 1260 series HPLC system (Agilent Technologies) with the modification that a tandem mass spectrometer QTRAP 6500 (SCIEX, Darmstadt, Germany) was used. Details of the instrument parameters and response factors for quantification can be found in S1 Table.

Indole-3-acetic acid was quantified using the same LC-MS/MS system with the same chromatographic conditions but using positive mode ionization with an ion spray voltage at 5500 eV. Multiple reaction monitoring (MRM) was used to monitor analyte parent ion → product ion fragmentations as follows: m/z 176 → 130 (collision energy (CE) 19 V; declustering potential (DP) 31 V) for IAA; $m/z$ 181 →133 + $m/z$ 181 →134 + $m/z$ 181 →135 (CE 19 V; DP 31 V) for D5-IES.

## Collection of phloem sap

Phloem sap was sampled applying centrifugation technique according to Hijaz and Killiny [88]. Briefly, the bark from young flush of M. domestica, P. communis and P. persica trees was manually removed with a clean scalpel and sliced into 2 cm pieces. After removing the bottom of a 0.5 ml Eppendorf tube, the tube was immersed in a second, larger tube (1.5 ml). Bark pieces were placed into the 0.5 ml tubes and centrifuged at 12,000 rpm at 4˚C for 10 min. Using this technique, we cannot exclude slight contamination from mesophyll cell content. As a possible slight contamination should be of minor importance, we refer to the samples as phloem sap throughout the publication. The extracted phloem sap was collected and the refractive index, the density and the viscosity were determined. Between measurements, sap samples were kept on ice to prevent evaporation of water and concentration of samples.

## Determination of the refractive index

The refractive index of the phloem sap was determined by the means of a handheld refractometer (type 45–81; Bellingham + Stanley Ltd., Tunbridge Wells, UK) and specified as ˚Brix. The refractometer was standardized for sucrose.

## Determination of the density of vascular saps (ρ)

The density of the phloem sap was measured using 0.5 µl glass capillaries (CAMAG, Muttenz, Switzerland) allowing the distinct determination of bulk and volume. The bulk of empty and filled capillaries was separately measured (Analytical Balance, Sartorius Weighing Technology GmbH, Göttingen, Germany) and subtracted to determine the pure mass of the phloem sap. Simultaneously, the corresponding volume inside the capillaries was calculated and the density was converted to gram per litre (g l⁻¹).

## Determination of the dynamic viscosity of vascular saps (η)

The measuring instruction of the dynamic viscosity was described in Adam et al. [89]. The dynamic viscosity of the phloem sap was quantified with 0.5 µl glass capillaries (CAMAG, Muttenz, Switzerland). The calibration required the determination of the specific viscometer constant ($\kappa$ = mPa l g⁻¹) by using $H_2O$ as reference solution with known dynamic viscosity ($\eta_0$ = 1.0087 mPa s) and density ($\rho_0$ = 997.9 g l⁻¹) for 20˚C. Following the measurement of elapsed time ($t_0$), the specific viscometer constant was calculated:

$$\kappa = \frac{\eta_0}{\rho_0\, t_0} \left( \frac{\text{mPa l}}{\text{g}} \right) \tag{3}$$

The dynamic viscosity of the phloem sap was calculated after the measurement of elapsed time (t) and determination of density ($\rho$):

$$\eta = \kappa * \rho * \text{t(mPa s)} \tag{4}$$

## Statistics

All statistical analyses were performed using R version 3.5.1 [90]. The data visualization was done with the package 'ggplot2' [91].

Mass flow, volumetric flow rate, morphology data and functional/physiological parameters: Linear mixed effect (LMM) or generalized linear mixed models (GLMM) were used to determine the effect of phytoplasma infection on phloem mass flow, volumetric flow rates and morphological and functional parameters (phloem sap viscosity and density) in *M. domestica*, *P. persica* and *P. communis* leaves. To account for non-independent errors, which may occur due to repeated measurements at each tree, trees were specified as a random factor in all models. Models with different error distributions and link-functions were compared by AICc (Akaike information criterion with correction for small sample size) with the *AICctab* function from the 'bbmle' package [92]. Models with the lowest AICc values were used if model assumptions were valid. LMMs were fitted with the *lmer* function from the 'lme4' package [93], and Type III analysis of variance (ANOVA) with Satterthwaite's method, which was calculated with the *Anova* function from the 'lmerTest' package [94]. GLMMs were fitted with the *glmer* function from the 'lme4' package, and Type II analysis of variance was calculated with the *Anova* function from the 'car' package [95] to determine treatment effects. Used error distribution, link-function and ANOVA results are specified in the S2–S4 Tables in the Supporting Information. A Mann–Whitney-U test was performed to compare the number of leaves of phytoplasma infected and uninfected fruit trees.

Phytohormone data and Brix values: Linear models were fitted to determine the influence of phytoplasma infections on the concentration of phytohormones and the relative density of phloem sap in *M. domestica*, *P. persica* and *P. communis* plants. In case of non-normality of the residuals the data was log, square root or box-cox transformed as specified in S4 and S6 Tables. Variance heterogeneity was detected in abscisic acid content in samples from *P. communis*. In this case the generalized least squares method (GLS) was applied with the *gls* function of the 'nlme' package [96]. The different variance in the treatments was incorporated into the model with the varIdent variance structure. Treatment effects were calculated by Type I analysis of variance.

Callose deposition: Linear models were fitted with the GLS method, to model the different variance structures of the data with the varIdent function. Treatment effects were calculated by Type I analysis of variance (S5 Table).

General procedure: For all models, the estimated marginal means (EMMs) and corresponding 95% confidence intervals were calculated and used to determine differences between treatment levels with the 'emmeans' package [97]. All model assumptions were validated graphically as recommended by Zuur *et al.* [98].

## Supporting information

**S1 Fig. Phytohormone concentrations of cis-12-oxo-phytodienoic acid (cis-OPDA) and indole-3-acetic acid (IAA) in uninfected and phytoplasma infected (a) apple, (b) pear and (c) peach trees.** Amounts of phytohormones were measured in the leaves of healthy and phytoplasma infected (a) apple, (b) pear and (c) peach. For apple, a virulent accession (3/6) was considered inducing apple proliferation (AP). Pear trees showed pear decline (PD) and peach trees showed the European stone fruit yellows (ESFY). Box-whisker plots with median as lines and jittered raw values as closed circles (corresponding to each measurement). Boxes represent

the interquartile range (IQR) and whiskers extend to 1.5*IQR. Bars represent the 95% confidence intervals with the estimated marginal means obtained from linear models as dots (both back transformed to the response scale). Letters indicate statistical differences between EMMs of groups at the 0.05 significance level.
(TIF)

**S2 Fig. Number of leaves of uninfected and phytoplasma infected apple, pear and peach trees.** For apple, a virulent accession (3/6) was considered inducing apple proliferation (AP). Pear trees showed pear decline (PD) and peach trees showed the European stone fruit yellows (ESFY). Box-whisker plots with median as lines and jittered raw values as closed circles (corresponding to each measurement). Boxes represent the interquartile range (IQR) and whiskers extend to 1.5*IQR. Bars represent the 95% confidence intervals with the estimated marginal means obtained from linear models as dots (both back transformed to the response scale). Letters indicate statistical differences between EMMs of groups at the 0.05 significance level.
(TIF)

**S1 Table. Details of analysis of phytohormones by LC-MS/MS [HPLC 1260 (Agilent Technologies)-QTRAP6500 (SCIEX)] in negative ionisation mode.** Mean phytohormone concentrations (ng gFM-1) extracted from leaf material of *Malus domestica*, *Pyrus communis* and *Prunus persica*. Diverse phytohormones were detected—salicylic acid (SA), jasmonic acid-iso leucine (JA-Ile), jasmonic acid (JA), abscisic acid (ABA), 12-Oxo-phytodienoic acid (cis-OPDA) and indole acetic acid (IAA). Different letters indicate significant differences between phytoplasma infected and uninfected trees compared among the species. AP = apple proliferation, PD = pear decline, ESFY = European stone fruit yellows, npl = amount plants, nl = amount leaves.
(DOCX)

**S2 Table. Specification and results of statistical models used for analysis of morphology parameters.** Length and width of leafs and diameters of midribs were measured from phytoplasma infected and non-infected *Malus domestica*, *Pyrus communis* and *Prunus persica*. The leaf ratio was calculated by dividing the leaf length by the leaf width, the midrib ratio is defined as the ratio between the midrib diameter and the leaf width. All parameters were compared between phytoplasma infected and non-infect trees within each plant species.
(DOCX)

**S3 Table. Specification and results of statistical models used for analysis of vascular morphology.** Areas of the vascular bundle, xylem, phloem and ten sieve elements were measured from phytoplasma infected and non-infected *Malus domestica*, *Pyrus communis* and *Prunus persica*. The ratio between xylem area and phloem area and of SE to phloem were calculated. All parameters were compared between phytoplasma infected and non-infect trees within each plant species.
(DOCX)

**S4 Table. Specification and results of statistical models used for analysis of the translocation in phloem sieve elements and physicochemical parameters.** The phloem mass flow velocity, the refractive index, the dynamic viscosity and density of phloem sap were measured from phytoplasma infected and non-infected *Malus domestica*, *Pyrus communis* and *Prunus persica*. The volumetric flow rate was calculated by multiplication of measured mass flow velocity with the median area of ten sieve elements. All parameters were compared between phytoplasma infected and non-infect trees within each plant species.
(DOCX)

**S5 Table. Specification and results of generalized least square models analysing the maximum callose fluorescence.** The callose deposition at the sieve plates was visualized by DAPI staining and fluorescence microscopy from phytoplasma infected and non-infected *Malus domestica*, *Pyrus communis* and *Prunus persica* and compared within each plant species. (DOCX)

**S6 Table. Specification and results of linear models used for analysis of phytohormone concentrations.** Concentrations of Phytohormones salicylic acid (SA), jasmonic acid-iso leucine (JA-Ile), jasmonic acid (JA), abscisic acid (ABA), 12-Oxo-phytodienoic acid (cis-OPDA) and indole acetic acid (IAA) were determined from phytoplasma infected and non-infected *Malus domestica*, *Pyrus communis* and *Prunus persica* leaves. Phytohormone analysed by liquid chromatography–mass spectrometry and concentrations compared between infected a non-infected trees within each plant species. (DOCX)

**S7 Table. Overview of mean phytohormone concentrations (ng gFM-1).** The phytohormone concentrations were determined for leaf material of several plant species—*Malus domestica*, *Pyrus communis* and *Prunus persica*. Diverse phytohormones were detected—salicylic acid (SA), jasmonic acid-iso leucine (JA-Ile), jasmonic acid (JA), abscisic acid (ABA), 12-Oxo-phytodienoic acid (cis-OPDA) and indole acetic acid (IAA). Different letters indicate significant differences between phytoplasma infected and uninfected trees compared among the species. AP = apple proliferation, PD = pear decline, ESFY = European stone fruit yellows, npl = amount plants, nl = amount leaves. (DOCX)

## Acknowledgments

We thank Sebastian Faus and Katharina Piwowarczyk (JKI, Dossenheim, Germany) for excellent assistance in the laboratory. We thank Felix Hergenhahn (JKI, Dossenheim, Germany) for grafting and cultivation of the plants. We thank Andrea Lehr (MPI for Chemical Ecology, Jena, Germany) for technical support. We are grateful to Dr. Eva Gross (Schriesheim, Germany) for language editing.

## Author Contributions

**Conceptualization:** Matthias R. Zimmermann, Jürgen Gross, Alexandra C. U. Furch.

**Data curation:** Jannicke Gallinger, Kerstin Zikeli, Matthias R. Zimmermann, Michael Reichelt, Alexandra C. U. Furch.

**Formal analysis:** Jannicke Gallinger, Kerstin Zikeli, Matthias R. Zimmermann, Alexandra C. U. Furch.

**Funding acquisition:** Jürgen Gross, Alexandra C. U. Furch.

**Investigation:** Jannicke Gallinger, Kerstin Zikeli, Matthias R. Zimmermann, Louisa M. Görg, Alexandra C. U. Furch.

**Methodology:** Jannicke Gallinger, Kerstin Zikeli, Matthias R. Zimmermann, Michael Reichelt, Alexandra C. U. Furch.

**Project administration:** Jürgen Gross, Alexandra C. U. Furch.

**Resources:** Axel Mithöfer, Jürgen Gross, Alexandra C. U. Furch.

**Software:** Jannicke Gallinger, Kerstin Zikeli, Matthias R. Zimmermann, Michael Reichelt, Alexandra C. U. Furch.

**Supervision:** Jürgen Gross, Alexandra C. U. Furch.

**Validation:** Matthias R. Zimmermann, Erich Seemüller, Jürgen Gross, Alexandra C. U. Furch.

**Visualization:** Jannicke Gallinger, Alexandra C. U. Furch.

**Writing – original draft:** Jannicke Gallinger, Kerstin Zikeli, Matthias R. Zimmermann, Jürgen Gross, Alexandra C. U. Furch.

**Writing – review & editing:** Jannicke Gallinger, Kerstin Zikeli, Matthias R. Zimmermann, Axel Mithöfer, Erich Seemüller, Jürgen Gross, Alexandra C. U. Furch.

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
