## [Decision Letter · Decision Letter 0]

5 Dec 2020

Dear Dr Furch,

Thank you very much for submitting your manuscript "Specialized 16SrX phytoplasmas induce diverse morphological and physiological changes in their respective fruit crops" for consideration at PLOS Pathogens. As with all papers reviewed by the journal, your manuscript was reviewed by members of the editorial board and by several independent reviewers. In light of the reviews (below this email), we would like to invite the resubmission of a significantly-revised version that takes into account the reviewers' comments.

We cannot make any decision about publication until we have seen the revised manuscript and your response to the reviewers' comments. Your revised manuscript is also likely to be sent to reviewers for further evaluation.

Sincerely,

Nian Wang

Guest Editor

PLOS Pathogens

Wenbo Ma

Section Editor

PLOS Pathogens

Kasturi Haldar

Editor-in-Chief

PLOS Pathogens

orcid.org/0000-0001-5065-158X

Michael Malim

Editor-in-Chief

PLOS Pathogens

orcid.org/0000-0002-7699-2064

Reviewer's Responses to Questions

**Part I - Summary**

Reviewer #1: The manuscript titled “Specialized 16SrX phytoplasmas induce diverse morphological and physiological changes in their respective fruit crops” submitted by Gallinger and colleagues describes physiological and morphological characterization of diseases caused by phytoplasmas 16SrX group in Rosaceae tree crops and suggests a link between the physiological responses to the infection and the differential survival between the crops.

The authors identified that apples, pears and peaches respond differently to the disease and suggested that apples, which have better survival rate, display several unique responses such as reduced leaf size, alteration of the leaf vascular morphology, increased number of leaves per tree, accumulation of stress-associated phytohormones and reduction in the velocity of phloem mass flow. In addition, callose accumulation in the vascular bundles is reduced in apples compared to peaches and pears. The authors suggested that these unique adaptations to the infection might partially explain why apples display higher tolerance to the disease compared to the other tree crops.

Diseases caused by fastidious phloem limited pathogens have devastating effects on many tree crops, but mechanistic understanding of such diseases is lacking because of the limitations in the study of these pathogens and tree crops. While the authors attempted to explain how the altered physiological responses to the bacteria in different tree crops affects disease manifestation, the data provided in the study is mainly descriptive and mechanistic understanding of these diseases was not supported by any experimental evidence. The experimental design of the study makes it difficult to understand whether the observations made are pathogen related or host related. In addition, sampling was limited to one time point and in some cases to a small number of leaves per plant which can significantly affect the understanding of disease progression.

Reviewer #2: Manuscript PPATHOGENS-D-20-02430 entitled ‘Specialized 16SrX phytoplasmas induce diverse morphological and physiological changes in their respective fruit crops’, is a new submission of the previously submitted manuscript PPATHOGENS-D-20-011286 by Jannicke Gallinger and coworkers. It provides a comparative description of the anatomical and physiological effects of three temperate fruit trees phytoplasmas of the taxonomic group 16SrX, namely “Candidatus Phytoplasma mali”, “Ca. P. pyri” and “Ca. P. prunorum” known as being respectively restricted to Malus, Pyrus and Prunus spp. The manuscript especially demonstrates that these plant species present contrasting responses to their specific specialized phytoplasma species. The manuscript shows that “Ca. P. mali” -infected apples trees exhibit reductions of average leaf surface associated with a reduced phloem mass flow velocity whereas leaf shape is not affected. In contrast, it is shown that Pyrus communis plants infected with “Ca. P. pyri” do not show such reduction of leaf size and have an increased phloem mass flow velocity or other physicochemical characteristics of phloem sap such as volumetric flow rate, dynamic viscosity and refractive index. They show that Prunus persicae plants infected by “Ca. P. prunorum” have yellowing leaves poorly affected in their size and surface but do not show significant change in their phloem mass flow velocity while they exhibit a significantly reduced volumetric flow rate. For both phytoplasma-infected Pyrus and Prunus the authors measure an increase of callose deposition. The phytohormone levels in Pyrus appeared unchanged whilst increase of salicylic acid and jasmonic acid-isoleucine are mostly found in Malus and Prunus.

According to me, the methodologies employed do not suffer flaws and future studies targeting germplasm resistance will benefit from the clarity of description of the parameters measured.

By comparison to the previous submission and in response to my comment the authors added data indicating the number of leaves (Lines 130-132 and 451-452; Tab 1, Tab 3 and S2) that show an increase in infected apple trees while it was unchanged in peach and pear trees.

I previously indicated that in my opinion the previous manuscript suffered a lack of discussion about the balance between the gain resulting of an increased tolerance in malus trees at the level of individuals and its cost at the level of the propagation of the phytoplasma in tree population. The authors have convincingly addressed this point by discussing it lines 322-343 of the discussion.

In response to my final comment :

“Finally, a response to infection can result from the intrinsic organogenetic structure of the host. In that case the response does not result from an adaptation but instead reflect the lack of adaptability of the host. I am not a specialist, but is there anything that differs in Apple trees vs Pyrus and Prunus regarding the vascular system morphogenesis ? If yes it should be discussed somewhere.”

The authors did not enrich the manuscript discussion. The authors however mentionned that Nii (1993) described differences in ingrowth structures in the sieve elements (SEs) of the phloem among single Rosacea species and found rather similar structures in Malus and Pyrus in comparison to Prunus (Nii, 1993) that was confirmed by Donghua and Xinzeng (1993) that also conclude about the length of the sieve elements in Prunus being shorter and connected to more companion cells (CCs) in comparison to Malus and Pyrus. In theory, the Prunus SEs have a higher flow resistance because of their shortness and the automatic increased number of sieve plates. But, that might be balanced with the higher number of CCs per SEs. The CCs guarantee the physiological function of the SEs and a rise of CCs per SEs might enable the Prunus species to establish higher pressures, or a higher viability/turnover, for instance.” Although I agree that the studies are scarce, it could be interesting for the reader to mention the above information somewhere.

Reviewer #3: The revised version of the manuscript submitted by Gallinger and colleagues is now convincing. The manuscript has been seriously revised. I thank the authors to have taken my previous remarks into account and to have responded to them either with a significant correction of the text or by convincing their point of view. Moreover, all the remarks made by the other reviewers also seem to have been taken into account. The entire manuscript is now in a less speculative form and the discussion thread is more result-based. The change of title seems appropriate. The results are described in a factual manner, and replaced in an ecological context. The last paragraph of the discussion remains open, and the population-wide impact is clearly envisioned. Most important in my opinion is the change in the form of the discussion which is more moderate with mention of alternative hypotheses.

In the present form, no doubt that this extensive work on characterization of changes induced by phytoplasmas on these three fruit trees will be of great interest for phytopathologists working with sap-limited bacteria. Future research directions are also interesting.

**Part II – Major Issues: Key Experiments Required for Acceptance**

Reviewer #1: 1. The study is almost solely descriptive. The only actively conducted experiment that was done in the study was inoculating the plants. While the information provided in this study is important and should be published it does not provide any evidence regarding disease mechanism and how or why the different crops respond differently to phytoplasma infections.

2. The distribution of phloem-limited bacteria and manifestation of disease is known to be uneven between different leaves in each tree. I therefore find the number of leaves per tree which were sampled for mass flow measurements and phytohormone analyses to be a bit low (two to four per plant).

Why the authors pooled only four leaves from each tree for phytohormone analyses? Did these leaves display similar symptoms? How the leaves were chosen for sampling (was it completely random?)? Have the authors distinguished between young and mature leaves?

3. The authors compared three different crops inoculated by three different 16SrX phytoplasma strains. I am assuming that these analyses were made this way because of restricted host range of each of the bacteria. With that in mind I still find the comparison between the diseases highly problematic since both the host and the pathogen are different in each system. That makes it difficult to determine whether the observed phenotypes are a result of differential response to the pathogen by the host or of differences in virulence between the different bacteria. This point was made during the previous review cycle of this manuscript and the authors claimed that phylogenetically, the bacterial strains are closely related to each other. I do not think this response properly addresses the reviewer’s comments.

Closely related species of various pathogens display high variation in virulence mechanisms, virulence intensity, host range and tissue specificity and therefore the assumption that phylogenetically related species will cause the same disease is incorrect.

4. According to the method section the sample timing was completely different for each crop. Pear, apple, and peach trees were infected with the bacteria for five, two and one years when samples were taken, respectively. I am also assuming (it is not stated in the text) that the trees were of different ages and were not kept in similar conditions and therefore were exposed to different biotic and abiotic stressors which affect their physiology.

If the authors aspire to compare between the hosts, they should inoculate and sample the plants at the same time and ideally collect several time points to assess how their physiological parameters change during disease progression.

5. Bacterial titers in the inoculated trees should be stated in the main text and not in the supplementary material.

Reviewer #2: I do not see key experiments that are absolutely required. In response to the request I made about a better description of the phytoplasma isolate used in this study, the authors provided the description of the isolates used in an additional table (table 4).

Reviewer #3: No key experiment required

**Part III – Minor Issues: Editorial and Data Presentation Modifications**

Reviewer #1: (No Response)

Reviewer #2: I found that the new proposed title better is more precise and does not mislead the reader as the previous one. All the comments I made about imprecise sentences or errors were addressed and all requested minor modifications have been made to the manuscript.

Table 4: “Source: PD infected C. pruni individuals caught in the …” what is the meaning of PD-infected ? Is it a mistake ? Maybe replace by ‘Ca. P. prunorum”-infected C. pruni ?

Reviewer #3: Lines 75-76 : « Phytoplasmas lacking many genes….fatty acid synthesis » seems incomplete. I guess that the authors were planning to link this sentence with the following one.

Lines 250-251 : I would suggest to add a reference after the statement ‘As the callose deposition in response to phytoplasma infections never results in a restriction of the bacteria’

Line 254 : it is not clear what are the “specific defense mechanisms” the authors are talking about here? I would suggest to be more specific.

Line 255-256 : I would suggest to remove the sentence ‘Thus, the apple-phytoplasma interaction might be older and better adapted to peach and pear’ as this idea is not clearly directly linked to the previous sentence.

Lines 269-271 : Although these data are interesting, I would suggest to remove both sentences at the end of this paragraph, as they correspond to unpublished results. I do not think that these data are essential here.

Lines 285-287 : A reference is lacking at the end of the statement ‘ Moreover, typical symptoms….imbalance of phytohormones’

Lines 328-329 : ‘might have emerged’ ?

PLOS authors have the option to publish the peer review history of their article (what does this mean?). If published, this will include your full peer review and any attached files.

Reviewer #1: No

Reviewer #2: No

Reviewer #3: No
---

## [Editor Report · Decision Letter 1]

7 Feb 2021

Dear Dr Furch,

Thank you very much for submitting your manuscript "Specialized 16SrX phytoplasmas induce diverse morphological and physiological changes in their respective fruit crops" for consideration at PLOS Pathogens. As with all papers reviewed by the journal, your manuscript was reviewed by members of the editorial board and by several independent reviewers. The reviewers appreciated the attention to an important topic. Based on the reviews, we are likely to accept this manuscript for publication, providing that you modify the manuscript according to the review recommendations.

This manuscript was not sent out for further reviews since two of the reviewers had minor issues previously, and the third reviewer declined to review the revision. Although some reviewers were concerned about “mechanistic” aspect of this study and the authors drew their conclusions from comparing three different bacterial strains in their different hosts (in which, each host was infected with only one bacterial strain). The editors decided that the overall discoveries outweigh the potential drawbacks.

Overall, the editors are happy with the revisions made in the current submission but suggest the authors to further revise the discussion by putting the discoveries in a broad context of phloem colonizing bacteria (e.g. Candidatus Liberibacter and Spiroplasma), rather than only focusing on Phytoplasmas. For instance, Candidatus Liberibacter also induces callose deposition and SA accumulation. Response of susceptible and tolerant citrus genotypes to Candidatus Liberibacter asiaticus have been investigated. Candidatus Liberibacter asiaticus utilizes SA hydroxylase and effectors to suppress plant defense. Plos pathogens aims to publish papers with broad impact. We hope the authors will be able to go beyond what have been stated about Phytoplasmas in the discussion.

Other minor comments:

Line 75: lacking should be ‘lack’

Line 269: “unknown specific defence mechanisms”. Please remove specific.

Table 5: change “unpublished to date” to “this study”

Reference 50 should be: Donghua Liu, Xinzeng Gao. Comparative anatomy of the secondary phloem of ten species of Rosaceae. IAWA J. 1993;14, 289-298.

Line 238: Donghua and Xinzeng (1993) should be “Liu and Gao (1993)”

Line 239: each SE iwa should be “each SE cell was”

Sincerely,

Nian Wang

Guest Editor

PLOS Pathogens

Wenbo Ma

Section Editor

PLOS Pathogens

Kasturi Haldar

Editor-in-Chief

PLOS Pathogens

orcid.org/0000-0001-5065-158X

Michael Malim

Editor-in-Chief

PLOS Pathogens

orcid.org/0000-0002-7699-2064
---

## [Editor Report · Decision Letter 2]

7 Mar 2021

Dear Dr Furch,

We are pleased to inform you that your manuscript 'Specialized 16SrX phytoplasmas induce diverse morphological and physiological changes in their respective fruit crops' has been provisionally accepted for publication in PLOS Pathogens.

Best regards,

Nian Wang

Guest Editor

PLOS Pathogens

Wenbo Ma

Section Editor

PLOS Pathogens

Kasturi Haldar

Editor-in-Chief

PLOS Pathogens

orcid.org/0000-0001-5065-158X

Michael Malim

Editor-in-Chief

PLOS Pathogens

orcid.org/0000-0002-7699-2064
---

## [Editor Report · Acceptance letter]

23 Mar 2021

Dear Dr Furch,

We are delighted to inform you that your manuscript, "Specialized 16SrX phytoplasmas induce diverse morphological and physiological changes in their respective fruit crops," has been formally accepted for publication in PLOS Pathogens.

Best regards,

Kasturi Haldar

Editor-in-Chief

PLOS Pathogens

orcid.org/0000-0001-5065-158X

Michael Malim

Editor-in-Chief

PLOS Pathogens

orcid.org/0000-0002-7699-2064